# Effect of miR-302b MicroRNA Inhibition on Chicken Primordial Germ Cell Proliferation and Apoptosis Rate

**DOI:** 10.3390/genes13010082

**Published:** 2021-12-28

**Authors:** Bence Lázár, Nikolett Tokodyné Szabadi, Mahek Anand, Roland Tóth, András Ecker, Martin Urbán, Maria Teresa Salinas Aponte, Ganna Stepanova, Zoltán Hegyi, László Homolya, Eszter Patakiné Várkonyi, Bertrand Pain, Elen Gócza

**Affiliations:** 1Animal Biotechnology Department, Institute of Genetics and Biotechnology, Hungarian University of Agriculture and Life Sciences, 2100 Godollo, Hungary; lazar.bence@uni-mate.hu (B.L.); Tokodyne.Szabadi.Nikolett@uni-mate.hu (N.T.S.); mahekanand@yahoo.com (M.A.); Toth.Roland.Imre@uni-mate.hu (R.T.); Ecker.Andras@phd.uni-mate.hu (A.E.); Urban.Martin@uni-mate.hu (M.U.); mtsalinasaponte@gmail.com (M.T.S.A.); 2Institute for Farm Animal Gene Conservation, National Centre for Biodiversity and Gene Conservation, 2100 Godollo, Hungary; varkonyi.eszter@nbgk.hu; 3Faculty of Medicine, Institute of Translational Medicine, Semmelweis University, 1089 Budapest, Hungary; gannastepanova2016@gmail.com; 4Institute of Enzymology, Research Centre for Natural Sciences, 1117 Budapest, Hungary; zotyahegyi@gmail.com (Z.H.); homolya.laszlo@ttk.hu (L.H.); 5Stem-Cell and Brain Research Institute, USC1361 INRA, U1208 INSERM, 69675 Bron, France; bertrand.pain@inserm.fr

**Keywords:** primordial germ cells, gga-miR-302b, miRNA inhibition, chicken, cell proliferation, apoptosis

## Abstract

The primordial germ cells (PGCs) are the precursors for both the oocytes and spermatogonia. Recently, a novel culture system was established for chicken PGCs, isolated from embryonic blood. The possibility of PGC long-term cultivation issues a new advance in germ cell preservation, biotechnology, and cell biology. We investigated the consequence of gga-miR-302b-5P (5P), gga-miR-302b-3P (3P) and dual inhibition (5P/3P) in two male and two female chicken PGC lines. In treated and control cell cultures, the cell number was calculated every four hours for three days by the XLS Imaging system. Comparing the cell number of control and treated lines on the first day, we found that male lines had a higher proliferation rate independently from the treatments. Compared to the untreated ones, the proliferation rate and the number of apoptotic cells were considerably reduced at gga-miR-302b-5P inhibition in all PGC lines on the third day of the cultivation. The control PGC lines showed a significantly higher proliferation rate than 3P inhibited lines on Day 3 in all PGC lines. Dual inhibition of gga-miR-302b mature miRNAs caused a slight reduction in proliferation rate, but the number of apoptotic cells increased dramatically. The information gathered by examining the factors affecting cell proliferation of PGCs can lead to new data in stem cell biology.

## 1. Introduction

Germ cell development generates totipotency throughout genetic as well as epigenetic regulation of the genome function. Primordial germ cells (PGCs) are the first germ cell population established during the development and serve as precursors for both the oocytes and spermatogonia [1]. PGCs, similarly to embryonic stem cells (ESCs) [2] and induced pluripotent stem cells (iPSCs) [3], are essential tools in stem cell biology and research. It is possible to genetically modify the PGCs and create germ cell chimeras from the transgenic PGCs, as was shown by Nakamura and co-workers [4]. Chicken PGCs can be easily isolated from the fertilized eggs at the early stages of embryonic development and can be maintained in vitro utilizing already established protocols [5,6]. This accessibility provides a unique tool to explore better medium formulations and investigate stem cell properties, such as factors governing pluripotency and self-renewal [7].

One of the emerged pluripotency factors in stem cell biology are miRNAs [8,9,10,11]. They play a fundamental role in maintaining the pluripotency of PGCs [12,13]. MicroRNAs (miRNAS) are small non-coding RNAs found in the genome that post-transcriptionally regulate gene expression via mRNA degradation or translation inhibition. The miRNAs are known to regulate important physiological and pathological processes [14]. The genes for the miRNAs are dispersed throughout the genome: some are intergenic, while others are located in intronic, or in exonic regions [11]. The biogenesis of miRNAs is a multi-step process in higher vertebrates. The process begins with RNA polymerase II transcribing these miRNA genes into primary-miRNA (pri-miRNA) transcipts. This pri-miRNA is further processed by the microprocessor complex consisting of DROSHA, DGCR8 and spliceosome components into precursor miRNA (pre-miRNA). The Exportin-5 enzyme exports pre-miRNAs out of the nucleus into the cytoplasm. This pre-miRNA is further cleaved by Dicer, another Ribonuclease III type enzyme, with the help of the TRBP into a miRNA duplex complex. The duplex consists of a 5P and a 3P strand, and the process of arm selection results in the guide RNA strand incorporating into the RNA-induced silencing complex (RISC), which mediates the RNAi-related gene silencing; the partial base pairing between the mature miRNA and the target mRNA leads to translation inhibition or mRNA degradation. Apart from this canonical maturation process, there are several Drosha- or Dicer-independent pathways of miRNA biogenesis [15,16,17]. MiRANs are crucial for gene regulation during pluripotency, self-renewal, and differentiation of ESCs and iPSCs [18]. They are expressed during the earliest embryonic developmental stages [9,19]. During embryonic development [19,20,21,22], the quantity of miRNAs is strongly regulated [16,20]. If the expression levels of non-coding RNAs become too high, they can act as cancer-promoting agents, which can cause the cells to escape standard mechanisms of control and become malignant in nature [8,17,23]. MicroRNAs belonging to the miR-302 family are emerging as key players in the control of cell growth. Khodayari and co-workers identified a novel mechanism of ephrin-A1 mediated anti-oncogenic signaling in malignant pleural mesothelioma (MPM) through miR-302b upregulation and inhibition of MM tumor sphere growth by inducing apoptosis [24]. Apoptosis is described by specific morphological and biochemical features in which caspase activation plays a central function as a component of both health and disease [25]. Understanding the mechanisms of apoptosis and other variants of programmed cell death provides deeper insight into various disease processes and may thus influence therapeutic strategy.

Our present study identified a novel effect of the gga-miR-302b on chicken PGC proliferation by inhibiting gga-miR-302b-5P (5P) and gga-miR-302b-3P (3P). We demonstrated that, in the case of gga-miR-302b-5P and gga-miR-302b-3P inhibition, PGCs showed a significantly reduced proliferation rate than the control PGCs. The inhibition of gga-miR-302b-5P reduced the apoptotic rate of all the examined PGC lines. Together, these findings suggest that the cells might undergo an additional pathway of cell degradation, besides apoptosis, that needs to be determined experimentally. The differentiation capacity of inhibited cells needs to be explored also.

## 2. Materials and Methods

### 2.1. Experimental Animals and Animal Care

Animals were kept according to the standard rules of the Hungarian Animal Protection Law (1998. XXVIII). Permission for experimental animal research at the National Centre for Biodiversity and Gene Conservation, Institute for Farm Animal Gene Conservation (Gödöllő, Hungary), was provided by the National Food Chain Safety Office, Animal Health and Animal Welfare Directorate (Budapest, Hungary). Fertilized eggs from Black Transylvanian Naked Neck Chicken were provided by the National Centre for Biodiversity and Gene Conservation, Institute for Farm Animal Gene Conservation (Gödöllő, Hungary).

### 2.2. Isolation, Establishment, and Maintenance of PGC Lines

Eggs were collected and incubated before the experiment. Circulating PGCs were isolated from fertilized eggs (HH stage 14–16) from Black Transylvanian Naked Neck chicken embryos and transferred to 300 μL culture medium in a 48-well plate without feeder cells. The culture medium was prepared as described by Whyte and colleagues [6]. The egg surface was cleaned up with 70% EtOH prior to opening. Next, 1 μL of blood was taken by a glass micro-pipette from the dorsal aorta of the embryo under a stereomicroscope. After 1–2 weeks, red blood cells died and PGCs prevailed. PGCs were cultured and used for the experiment. Half of the medium was changed every other day. When the total cell number reached 1.0 × 10^5^, the cells were divided into two and propagated at 2–4.0 × 10^5^ cells/mL medium in a 24-well plate [26]. Tissue samples for sex-determination were collected from every isolated embryo and stored at −20 °C until further use. The time of isolation, the exact age of the embryos (HH stages) cells [27], and visual the presence/absence of developmental abnormalities were recorded.

### 2.3. Cell Counting Using Arthur Fluorescence Cell Counter

The cell counting of the chicken PGCs before preparing the cell proliferation assay was performed using the Arthur Novel Fluorescence Cell Counter (NanoEnTek, Pleasanton, CA, USA). Two separate counts in parallel confirmed the results. For each line, PGCs were collected from 6 wells of a 24-well plate. Cell concentration was calculated from 25 μL PGC suspension by the Arthur Cell Counter. After cell number calculation, 1 × 10^3^ cells were placed into each well of one 96-well plate. Further, the cell number was measured every 4 h, for three days, using High-Content Screening Molecular Device. The measurement using ImageXpress Micro XLS Imaging System with a built-in incubator was performed at the Molecular Cell Biology Research Group (Institute of Enzymology, Research Centre for Natural Sciences): 96-well plate, with 6-6 parallel wells, of four previously established PG cell lines: two male (M1: #508-ZZ; M2: #512-ZZ) and two females (F1: #509-ZW; F2: #513-ZW) PGC line prepared for each condition. At the end of the experiment, the samples were collected for apoptosis measurement and immunostaining (Appendix A).

### 2.4. Apoptosis Rate Measurement Using Arthur Fluorescence Cell Counter

Apoptosis rate was measured on Day 3. 10× Annexin V Binding Buffer was diluted to 1× (AD10—Dojindo Molecular Technologies, Inc., Rockville, MD, USA) with tissue culture grade water (Applied Biosystems, Life Technologies, Carlsbad, CA, USA). Both the Annexin V FITC and the Propidium Iodide components were used at 5 μL/test in a final volume of 100 μL. The apoptotic and necrotic rate of the PGCs was measured using the Arthur Novel Fluorescence Cell Counter (NanoEnTek, Pleasanton, CA, USA).

### 2.5. Immunostaining of PGCs

PGCs were fixed with 4% PFA for 10 min. After washing with PBS, the fixed cells were blocked for 45 min with a blocking buffer containing 5% (*v*/*v*) BSA, then were incubated with each of the primary antibodies including mouse anti-SSEA-1 (1:10, Developmental Studies Hybridoma Bank, Iowa City, Iowa, USA) and rabbit anti-VASA (CVH) (1:1000; kindly provided by Bertrand Pain, Lyon, France). After overnight incubation in the primary antibody solution in a humid chamber at 4 °C, the cells were washed three times with PBS. Then the cells were incubated for 1 h with the secondary antibodies, Alexa-Fluor488-conjugated donkey anti-rabbit-IgG (Applied Biosystems, Life Technologies, Carlsbad, CA, USA) or Alexa-Fluor-555-conjugated donkey-anti-mouse-IgM (Applied Biosystems, Life Technologies, Carlsbad, CA, USA) in a dark, humid chamber, at 37 °C temperature. After washing once with 1× PBS, the nucleus was stained with TO-PRO^®^-3 stain for 15 min (1:500, Molecular Probes Inc., Eugene, OR, USA). After three rounds of 1x PBS, wash coverslips were mounted on the slide with the application of 20 μL VECTASHIELD^®^ Mounting Media (Vector Laboratories Inc., Burlingame, CA, USA) and analysed by confocal microscopy (TCS SP8, Leica Microsystems IR GmbH Wetzlar, Hessen, Germany). Negative controls were stained only with the secondary antibodies.

### 2.6. RNA Isolation

We collected the cells for RNA isolation in RNA Aqueous Lysis Buffer Micro Kit (Applied Biosystems). The isolated RNA was then used for qRT-PCR analysis for stem cell- and germ cell-specific markers as well as for the miRNAs: gga-miR-302b-5P and gga-miR-302b-3P (Table 1 and Table 2). The concentration of the extracted RNAs was determined using the NanoDrop One Spectrophotometer (Thermo Fisher Scientific, Waltham, MA, USA). The isolated RNA samples were stored at −70 °C.

### 2.7. CDNA Writing, qRT-PCR

The extracted RNA samples were reverse transcribed into cDNA with High-Capacity cDNA Reverse Transcription Kit, following the instructions of the manufacturer (Applied Bio systems, Life Technologies, Carlsbad, CA, USA). RT master mix was used for cDNA writing. The cDNA was stored at −20 °C. The synthesized cDNA was then used for qRT-PC. The reaction was performed by Eppendorf MasterCycler Realplex machine. TaqMan PCR master mix was applied for the qPCR as a double-stranded fluorescent DNA-specific dye according to the manufacturer’s instructions (Applied Bio systems, Life Technologies, Carlsbad, CA, USA). GAPDH was used as an internal control (housekeeping gene). For each gene examined, three parallels were analysed, fluorescence emission was detected and relative quantification was calculated with the GenEx7 program (MultiD Analyses AB, Göteborg, Sweden).

The qRT-PCR analysis was used for checking the expression of the PGC-specific and stem cell-specific markers in the chicken PGCs. The mRNA analysis was done for the following genes given in Table 1 below.

The qRT-PCR analysis was done to check the expression of miR-302b-3P and miR-302b-5P mature miRNA expression. MiR-92 was used as an internal control (Table 2).

### 2.8. Inhibition gga-miR-302b-5P and gga-miR-302b-3P Using MicroRNA Inhibitors

Cultured chicken PGCs were transfected with inhibitors against the gga-miR-302b-5P and gga-miR-302b-3P at 100 nM final concentration using the transfection agent siPORT (Applied Biosystems, Life Technologies, Carlsbad, CA, USA). We used the following inhibitors given in Table 3 below.

### 2.9. Statistical Analysis

The expression or repression of the target gene relative to the internal control gene in each sample was calculated with the GenEx 7.0 program (Multiday, SE) using the formula 2^−ΔΔCt^, where ΔCt = Ct target gene—Ct internal control and ΔΔCt = ΔCt test sample—ΔCt control sample. Statistical differences between the examined groups were assessed by *t*-test using the GenEx 7.0 software. The data are presented as mean ± SD and *p* values of less than 0.05 were regarded as statistically significant. Labels on the pictures: *p* < 0.05 *, *p* < 0.01 **, *p* < 0.001 ***.

## 3. Results

There were two male (M1: #508-ZZ; M2: #512-ZZ) and two female (F1: #509-ZW; F2: #513-ZW) PG cell lines used in this study. The cell number of these cell lines was measured every 4 h for three days, using the XLS Imaging system with a built-in incubator. Our aim was to examine the proliferation rate of the PGCs on Day 1, Day 2, and Day 3 after the inhibition of gga-miR-302b-5P (5P) or gga-miR-302b-3P (3P) or using anti-gga-miR-302b-5P and anti-gga-miR-302b-3P inhibitors together (5P/3P inhibition). For example, the proliferation rate on the third day was calculated by dividing the average cell number counted on the third day (h76) by the second day (h52). We compared the proliferation rate of control and treated lines on the first day (28 h/4 h), on the second day (52 h/28 h) and on the third day (76 h/52 h) (Appendix A).

### 3.1. Characterization of Stem Cell- and Germ Cell-Specific Marker Expression Profile of PG Cell Lines at Day 0

The used PG cell lines were characterized before the experiments.

Expression of CVH and POUV were measured. We found higher expression of CVH (germ cell-specific marker, Figure 1a) in male lines (M1, M2) than in females (F1, F2). POUV (stem cell-specific marker, Figure 1b) expression was the highest in the case of the M1 PGC line. Furthermore, a significant difference was found between the proliferation rate of male and female lines. In the case of males, we found a significantly higher proliferation rate compared to female lines (Figure 1c). Gga-miR-302b-5P (Figure 1d) and gga-miR-302b-3P (Figure 1e) miRNAs were also analysed at Day 0. The expression of both miRNAs was the highest in the case of cell line M2, but the difference was not significant. Interestingly, the 5P/3P ratio was the highest in cell line F2 (Figure 1f); although that was not the case regarding the proliferation rate (Figure 1c).

The immunostaining was done to analyse the expression of stem cell-specific SSEA-1 and germ cell-specific CVH. We found high expression for examined markers in all cell lines. Aggregation of cells in the case of the female lines (F1, F2) was also detectable (Figure 2).

The cell number was increased in all cell lines during the cultivation period (4–76 h, Figure 3B). The M1 and M2 (male, ZZ genotypes) lines showed a significantly decreased proliferation rate on Day 2 and Day 3 compared to Day 1 (Figure 3A(a,b)). The proliferation rate of F1 (female, ZW genotype, Figure 3A(c)) PGCs did not change, while the proliferation rate of F2 (female, ZW genotype, Figure 3A(d)) line increased significantly on Day 2 and Day 3 compared to Day 1. This outcome can explain why we got the highest 5P/3P ratio in the case of F2 PGCs (Figure 1f). This finding fit with our earlier result [28] also. The immunostaining revealed high SSEA-1 and CVH expression in all lines (Figure 3A(e–h)).

### 3.2. Effect of Anti-gga-miR-302b-5P, -3P and 5P/3P Inhibition on the Proliferation Rate of PGC Lines

We aimed to examine the proliferation rate of the PGCs on Day 1, Day 2 and Day 3 after the inhibition of gga-miR-302b-5P or gga-miR-302b-3P or the inhibition of both (Appendix A).

In the case of the M1 cell line (Figure 4a), on Day 2, the proliferation rate of 5P inhibited cells was significantly lower compared to the control. On Day 3, both 5P and 3P inhibited lines showed lower proliferation rates. The dual inhibition did not cause any detectable differences. We fixed the cells after three days of cultivation. Immunostaining was performed to detect the expression of SSEA-1 (stem cell-specific cell surface marker) and CVH (germ cell-specific marker expressing in the cytoplasm) (Figure 4b–d). In the case of 5P inhibition (Figure 4b), cytoplasm blebbing on most of the cell surface was visible, while in the control condition on Day 0 and Day 3, we could not find membrane deformities (Appendix A). The staining of these protrusions showed that they contain only cytoplasm. No sign of nuclear degradation was visible.

Cell line M2 (ZZ genotype) (Figure 5a) showed a significant proliferation rate increase compared to the control after inhibition of 3P and 5P/3P on Day 1. On Day 2, no difference was observed between the treated and the control groups. On Day 3, all three inhibitions caused the proliferation to slow down. Small protrusions of the cytoplasm were present only after inhibiting against the 5P (Figure 5b–d).

Cell line F1 (ZW genotype) (Figure 6a) showed a significant proliferation rate increase compared to the control after inhibition of 5P and 3P on Day 1. On Day 2, also significant proliferation rate increase was observed, but in the case of 3P and 5P/3P inhibition. Then, on Day 3, all three inhibitions caused the proliferation to slow down. Small protrusions of the cytoplasm were present after all inhibition treatments (Figure 6b–d).

Cell line F2 (ZW genotype) (Figure 7a) showed a significant proliferation rate increase compared to the control after 3P and 5P/3P inhibition on Day 1. On Day 2, no difference was observed between the treated and the control groups. Then, on Day 3, all three inhibitions caused the proliferation to slow down. Small protrusions of the cytoplasm were present after 3P and 5P/3P inhibition (Figure 7c,d).

Based on our results, we can conclude that, on the third day of the experiment, inhibition of 5P and 3P caused a drastic decrease in the proliferation rate for all cell lines compared to the control (Figure 8); however, the time scale of inhibition was different in the cell lines. Figure 8 shows the proliferation rate comparison on Day 3. It is visible that the control samples had significantly higher proliferation rates in the case of M2, F1, F2 (Figure 8b–d) than the treated ones. M1 PGCs did not show a difference compared to control after 5P/3P inhibition (Figure 8a).

It is also shown by the immunostaining that the 5P inhibited lines had a higher amount of blebbing cells than the other conditions (Figure 8e–h).

### 3.3. Determination of the Apoptotic, Late Apoptotic, and Necrotic Cell Ratio of PGC Lines

The apoptotic/necrotic staining was done with the Apoptotic Cell Detection Kit. The cell number calculation was performed using the Arthur Novel Fluorescent Cell Counter. We used two parallel measurements. The average apoptotic, late apoptotic, and necrotic ratios are illustrated in Figure 9. We found that 5P-inhibited PG cells showed a significantly lower apoptotic rate in all cell lines. The 5P/3P inhibition caused the highest percentage of cell death. The highest apoptotic rates were found in the case of the F1 cell line.

## 4. Discussion

Chicken Primordial germ cells (PGCs) can be found in the central region of the forming embryo disc. They are the first germ cell population during the development. These cells are precursors of oocytes and spermatogonia. Our study investigated the effect of miR-302b-5P and -3P on cell proliferation and apoptosis via a miRNA inhibition-based assay in PGCs. The information collected by examining the factors affecting cell proliferation of PGCs can lead to new data in stem cell biology [29].

It was published that Lin28 is a negative regulator of let-7 miRNA, and it is essential for PGC development in mouse [30]. The miR-290-295 and miR-17-92 clusters are important in mouse PGCs. MiR-290-295 cluster deficiency in mice leads to embryo lethality and germ cell deficiencies together with PGC migration problems [31]. Overexpression of Lin28 is associated with human germ-cell tumors [30]. In chicken, miR-363 is involved in gonadal development [3]. Moreover, miR-181a inhibits PGC differentiation [32].

The miR-302 cluster is embryonic stem cell-specific and evolutionarily conserved in vertebrates. The miR-302/367 cluster, generally consisting of five members, miR-367, miR-302d, miR-302a, miR-302c and miR-302b, is ubiquitously distributed in vertebrates and occupies an intragenic cluster located in the gene La-related protein 7 (LARP7) [32]. The cluster is cited for playing vital roles in diverse biological processes, such as the pluripotency of human embryonic stem cells (hESCs), self-renewal and reprogramming [28].

The miR-302-367 promoter is known to be transcriptionally regulated by the ESC-specific transcription factors Oct3/4, Sox2 and Nanog, and its activity is restricted to the ESC compartment. Functionally, this cluster regulates the cell cycle in ESCs, promoting self-renewal and pluripotency, therefore representing a master regulator in maintaining hESC stemness. It helps overcome the G1 to S phase transition during the cell cycle.

It was observed that miR-302 is endogenously highly expressed in human embryonic stem cells (hESCs) and human-induced pluripotent stem cells (hiPSCs). Inhibition of miR-302 using antagomirs resulted in downregulation of the self-renewal rate of hESCs, hiPSCs which was observed via cell colony formation assay [33].

We reported earlier a concordant dysregulation between the two arms of gga-miR-302b-5P and 3P [28] In this study, we identify that inhibition of the 3P arm slightly decreased the proliferation, indicating a role of the miR-302b-3P arm in cell proliferation. The cells inhibited with the 5P arm had a lower apoptotic rate than the cells that showed a dual inhibition. This result is consistent with Wu et al. [33], where the 5P acts as a proliferation promoter and oncomiR. Inhibition of miR-302b 5P arm decreased the proliferation rate and lowered the apoptotic rate, indicating a role of the 5P arm in promoting the proliferation of PGCs in vitro and in vivo. Our experimental data have confirmed the proliferation rate reduction after the inhibition of gga-miR-302b-5P and gga-miR-302b-3P (Figure 10).

It has been demonstrated that the members of the miR-302/367 cluster have a critical role in regulating the balance of G1-to-S transition. Three target genes were identified: CDK2, Cyclin D1/D2 and BMI-1 [34].

**Figure 10 genes-13-00082-f010:**
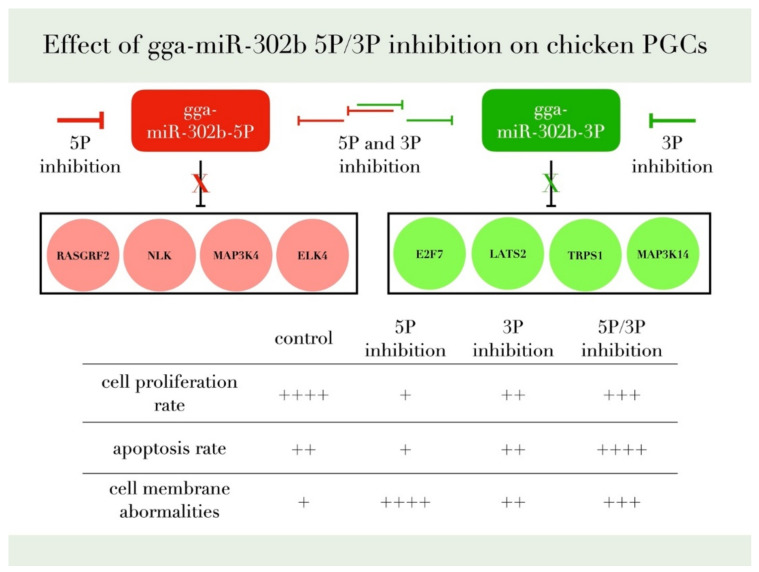
This figure summarizes our findings and the most relevant target genes of gga-miR-302b-5P and gga-miR-302b-3P predicted by the bioinformatic analysis published previously [28,35,36].

It is also essential to identify the miR-302b-5P and -3P expression pattern. In nature, both the 5P or 3P miRNA form has been reported, and 5P/3P ratio depends on temporal, spatial, physiological, and pathological conditions. Specific arm selection is supposed to be thermodynamically controlled. Changes in strand selection in cancer cells or developmental stages are possibly associated with the presence of signals. Dicer may affect the 5P/3P strand selection of concurrent expression in cancer cells. In summary, co-regulation of the 5P/3P miRNA in normal and pluripotent tissues and cells is subjected to subtle physiological changes in pre-miRNA processing enzymes and signals [29,37].

It was issued previously that miR-302b has an essential role in cellular glucose metabolism. Glucose energy and metabolism are crucial for embryonic stem cells, induced pluripotent stem cells, and germ cells. A variety of miRNAs regulate glucose metabolism in the pancreas, liver, brain, and muscle adipose tissue. Research papers published by Rengaraj and colleagues [12,32,38,39] described that miR-302b has a significant regulatory effect on glucose phosphate isomerase (GPI), which plays an essential role in glucose metabolism. Alteration of GPI can be associated with the abnormal function of stem and germ cells.

Oct4 and Sox2 are transcription factors essential for pluripotency during early embryogenesis and maintaining embryonic stem cell (ESC) pluripotency. They bind to a conserved promoter region of miR-302 [40]. MiR-302a is predicted to target many cell cycle regulators. Moreover, miR-302a represses the translation of cyclin D1 in hESCs. The transcriptional activation of miR-302 and the translational repression of its targets, such as cyclin D1, may provide a link between Oct4/Sox2 and cell cycle regulation in pluripotent cells. Varying stability and stoichiometry of such complexes offer a means to fine-tune developmental decisions [41].

The apoptotic rate of the cells inhibited against gga-miR-302b-5P showed a negative correlation, while the proliferation rate still dropped. Apoptosis occurs whenever there is an injury to the cell that cannot be repaired. The most common cause of such injury is DNA damage. Whenever cells detect DNA damage, they trigger the response of the p53 gene, which is the most crucial inducer of apoptosis. The second pathological condition under which apoptosis can be observed is whenever misfolded protein accumulates [42]. Multiple types of death can be observed simultaneously in tissues or cell cultures exposed to the same stimulus [43]. The externalization of phosphodiesterase is an early event of apoptosis, occurring while the plasma membrane remains intact. The biological spectrum of cell deaths is much more diverse. The formation of additional “blebbing” on the membrane surface of inhibited cells needs to be addressed. The non-apoptotic membrane blebbing is known to be a cellular migration mechanism [44].

We found that inhibition of miRNA gga-miR-302b-5P has a dual effect. It was reported that the miR-302b acts as an anti-tumor specific miRNA. It has been stated that various miRNAs regulate the intrinsic and the extrinsic pathway for apoptosis in cancer cells. The role of miR-302b-5P in hepatocellular carcinoma (HCC) is still unclear [33]. Guo and co-workers published that upregulation of the miRNAs leads to extensive upregulation of pro-apoptosis genes and pathways, leading to extensive cell death and blebbing [45]. These results indicate the tumor suppressor role of miR-302b-3P in the pathogenesis of gastric cancer. MiR-302b-3p promotes self-renewal properties in Leukemia Inhibitory Factor-Withdrawn Embryonic Stem Cells [46]. miR-302b exhibited anti-tumor activity by reversing EphA2 regulation, which relayed a signalling transduction cascade that attenuated the functions of N-cadherin, β-catenin, and Snail (markers of Wnt/β-catenin and epithelial-mesenchymal transition, EMT). This modulation of EphA2 also had distinct effects on cell proliferation and migration in vivo [47].

MAPK signalling is responsible for maintaining pluripotency and proliferation in mammals. It can be hypothesized that probably the high expression of gga-miR-302b-5P is contributed to controlling the MAPK signaling pathway [21] and TGFβR2. Their targets are downstream targets in other molecular pathways like p53 signalling, FOXO signaling, TGFβ signalling, and apoptosis. MiR-302b-5P might cause a high proliferation rate in PGC lines through inhibiting the MAPK pathway components.

## 5. Conclusions

Our findings could help to improve the in vitro conditions for PGC cultivation to gain stable germline competence comparable to in vivo PGCs. Many proteins and ligands are targets of the gga-miR-302b-5P and -3P. We can hypothesize that the inhibition of the 5P arms of miR-302b might lead to an upregulation of these pathways, causing a decrease in proliferation and apoptosis rate. Investigating the cell cycle regulatory function of gga-miR-302b-3P can help to understand its tumor suppressor role of miR-302b-3P in different types of cancer.

## Figures and Tables

**Figure 1 genes-13-00082-f001:**
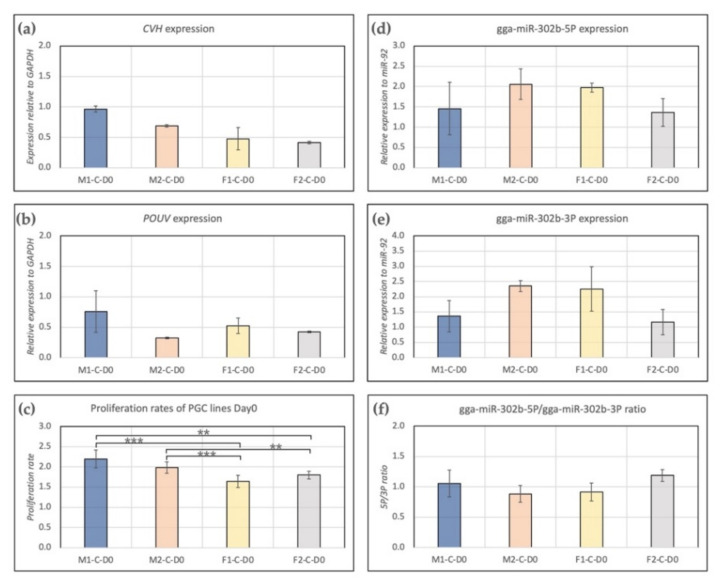
Characterization of PGC lines M1, M2, F1 and F2 on Day 0 of the experiments. Expression of CVH (**a**), POUV (**b**), gga-miR-302b-5P (**d**) and gga-miR-302b-3P (**e**) were analysed. GAPDH (in the case of CVH and POUV) and miR-92 (in the case of microRNAs) were used as housekeeping genes in the experiments. The proliferation rate (**c**) and 5P/3P ratio (**f**) for the PGC lines are also presented (*p* < 0.01 **, *p* < 0.001 ***).

**Figure 2 genes-13-00082-f002:**
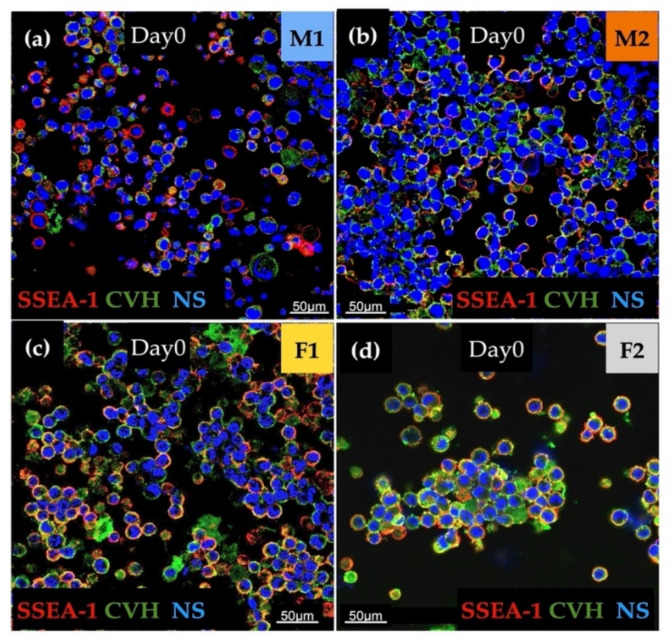
The immunostaining was performed with SSEA-1 (red), CVH (green) and TO-PRO™-3 for nuclear staining (blue). We examined M1 (**a**), M2 (**b**), F1 (**c**) and F2 (**d**) PG cell lines (Scale: 50 µm).

**Figure 3 genes-13-00082-f003:**
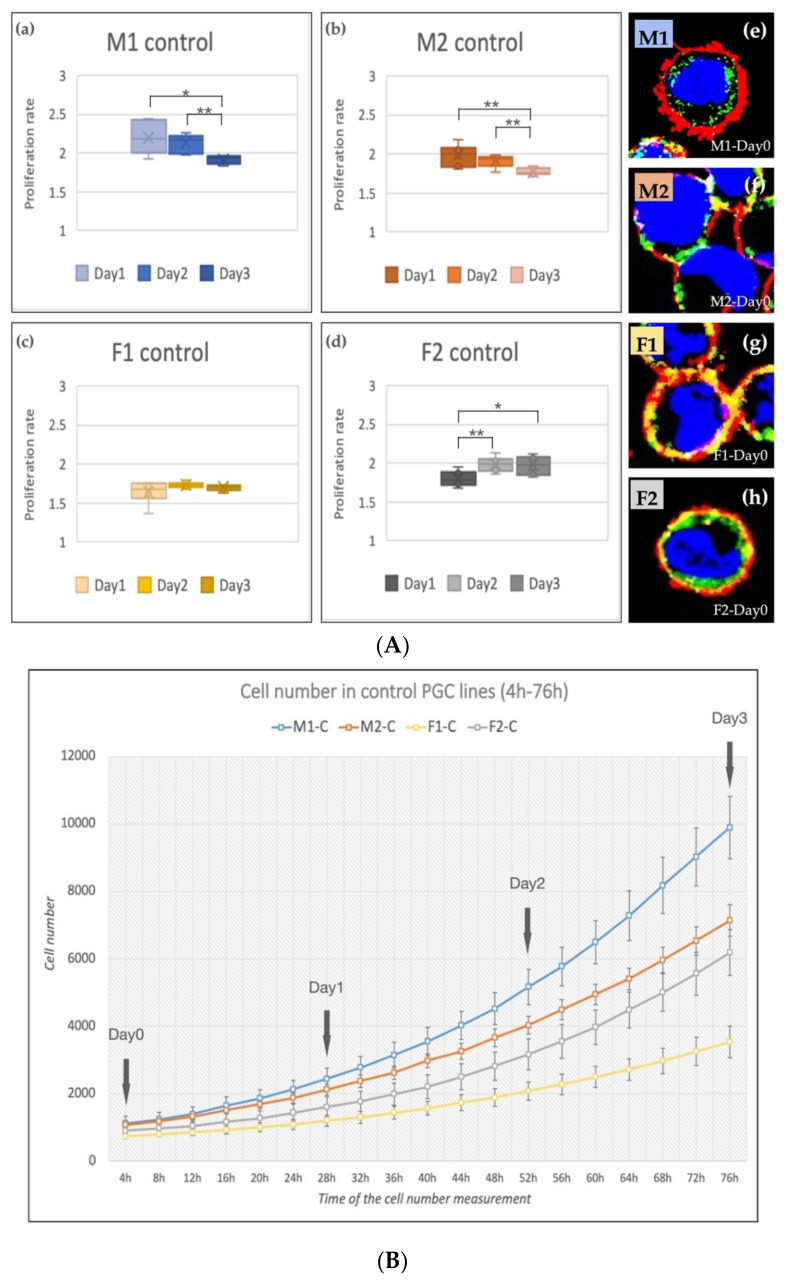
(**A**) The proliferation rate of the control PGC lines on Day 1, Day 2 and Day 3 (**a**–**d**) and immunostaining (**e**–**h**) of the four lines are shown in the figure. The immunostaining was achieved with SSEA-1 (red), CVH (green) and TO-PRO™-3 for nuclear staining (blue). (**B**) Analysis of the cell number. The cell number measurements were performed every 4 h (from 4 h to 76 h). We calculated the proliferation rates on Day 1, Day 2 and Day 3 (*p* < 0.05 *, *p* < 0.01 **).

**Figure 4 genes-13-00082-f004:**
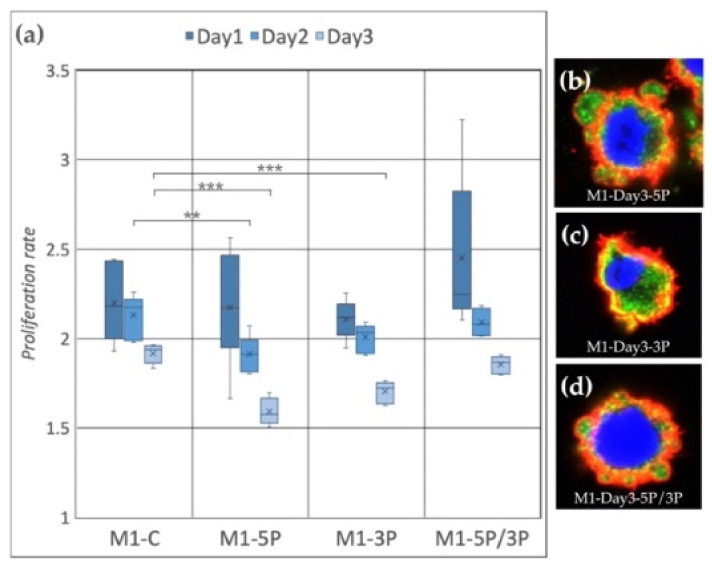
(**a**) Proliferation rate of M1 (ZZ genotype) PGC line on Day 1, Day 2 and Day 3. M1-C: non-inhibited, M1-5P: gga-miR-302b-5P inhibition, M1-3P: gga-miR-302b-3P inhibition, M1-5P/3P: gga-miR-302b-5P and gga-miR-302b-3P inhibition. (**b**–**d**) Immunostaining of M1 PGCs after three days of cultivation in different culture conditions. The immunostaining was achieved with SSEA-1 (red), CVH (green) and TO-PRO™-3 for nuclear staining (blue) (*p* < 0.01 **, *p* < 0.001 ***).

**Figure 5 genes-13-00082-f005:**
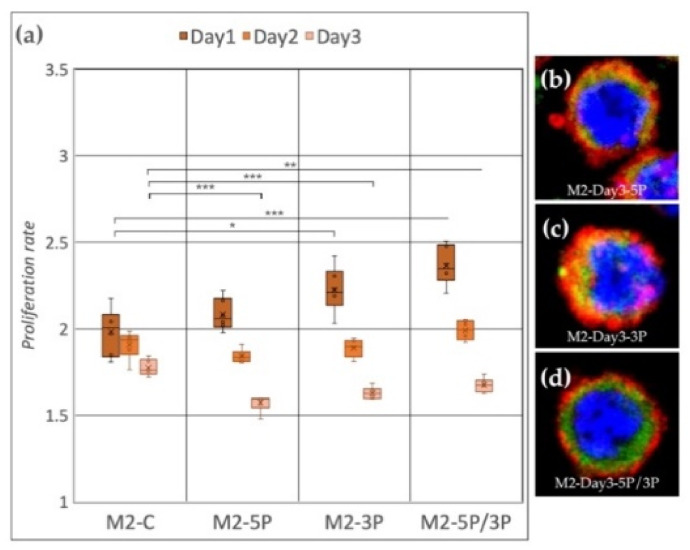
(**a**) Proliferation rate of M2 (ZZ genotype) PGC line on Day 1, Day 2 and Day 3. M2-C: non-inhibited, M2-5P: gga-miR-302b-5P inhibition, M2-3P: gga-miR-302b-3P inhibition, M2-5P/3P: gga-miR-302b-5P and gga-miR-302b-3P inhibition. (**b**–**d**) Immunostaining of M2 PGCs after three days of cultivation with different culture conditions. The immunostaining was achieved with SSEA-1 (red), CVH (green) and TO-PRO™-3 for nuclear staining (blue) (*p* < 0.05 *, *p* < 0.01 **, *p* < 0.001 ***).

**Figure 6 genes-13-00082-f006:**
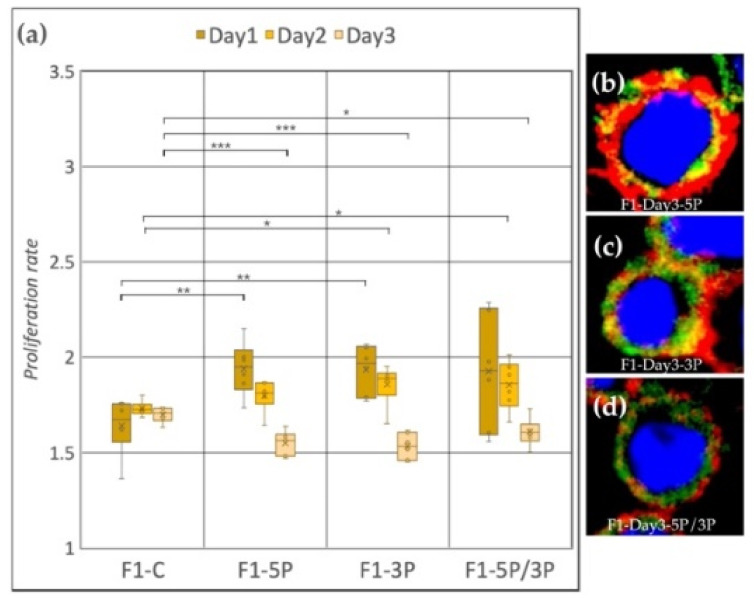
(**a**) Proliferation rate of F1 (ZW genotype) PGC line on Day 1, Day 2 and Day 3. F1-C: non-inhibited, F1-5P: gga-miR-302b-5P inhibition, F1-3P: gga-miR-302b-3P inhibition, F1-5P/3P: gga-miR-302b-5P and gga-miR-302b-3P inhibition. (**b**–**d**) Immunostaining of F1 PGCs after three days of cultivation with different culture conditions. The immunostaining was achieved with SSEA-1 (red), CVH (green) and TO-PRO™-3 for nuclear staining (blue) (*p* < 0.05 *, *p* < 0.01 **, *p* < 0.001 ***).

**Figure 7 genes-13-00082-f007:**
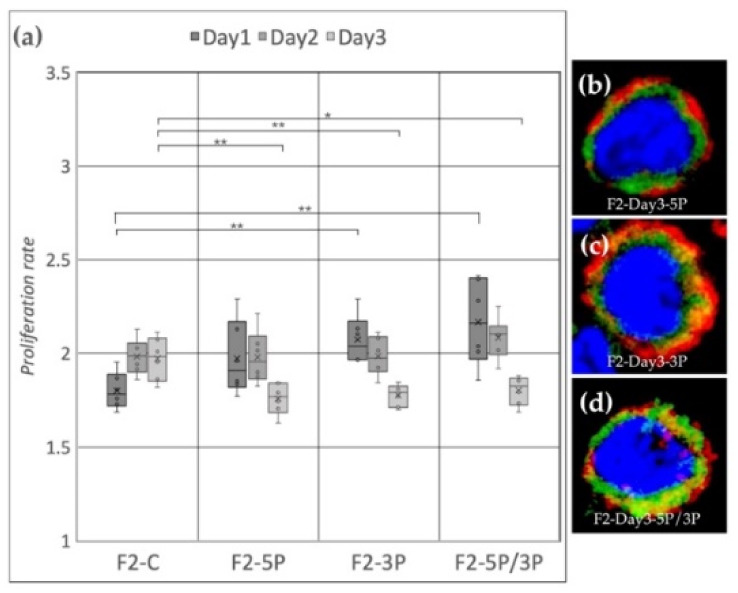
(**a**) Proliferation rate of F2 (ZW genotype) PGC line on Day 1, Day 2 and Day 3. F2-C: non-inhibited, F2-5P: gga-miR-302b-5P inhibition, F2-3P: gga-miR-302b-3P inhibition, F2-5P/3P: gga-miR-302b-5P and gga-miR-302b-3P inhibition. (**b**–**d**) Immunostaining of F2 PGCs after three days of cultivation with different culture conditions. The immunostaining was achieved with SSEA-1 (red), CVH (green) and TO-PRO™-3 for nuclear staining (blue) (*p* < 0.05 *, *p* < 0.01 **).

**Figure 8 genes-13-00082-f008:**
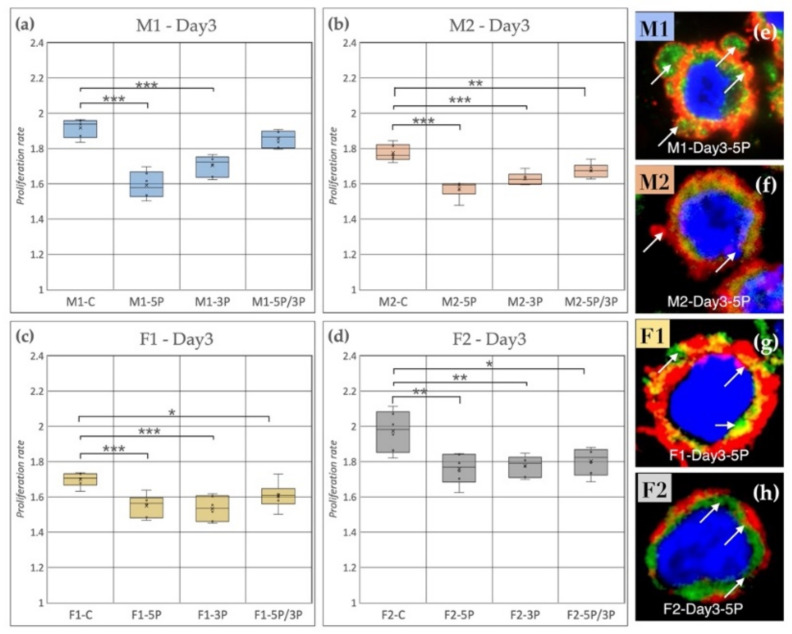
(**a**) Comparison of the proliferation rate of M1 (**a**), M2 (**b**) (ZZ genotype) and F1 (**c**), F2 (**d**) (ZW genotype) PGC lines on Day 3. C: non-inhibited, 5P: gga-miR-302b-5P inhibition, 3P: gga-miR-302b-3P inhibition, 5P/3P: gga-miR-302b-5P and gga-miR-302b-3P inhibition. (**e**–**h**) Immunostaining of M1, M2, F1 and F2 PGCs after three days of cultivation in anti-gga-miR-302b-5P inhibitor-containing medium. The immunostaining was achieved with SSEA-1 (red), CVH (green) and TO-PRO™-3 for nuclear staining (blue). Arrows show the blebbings on the cell surface (*p* < 0.05 *, *p* < 0.01 **, *p* < 0.001 ***).

**Figure 9 genes-13-00082-f009:**
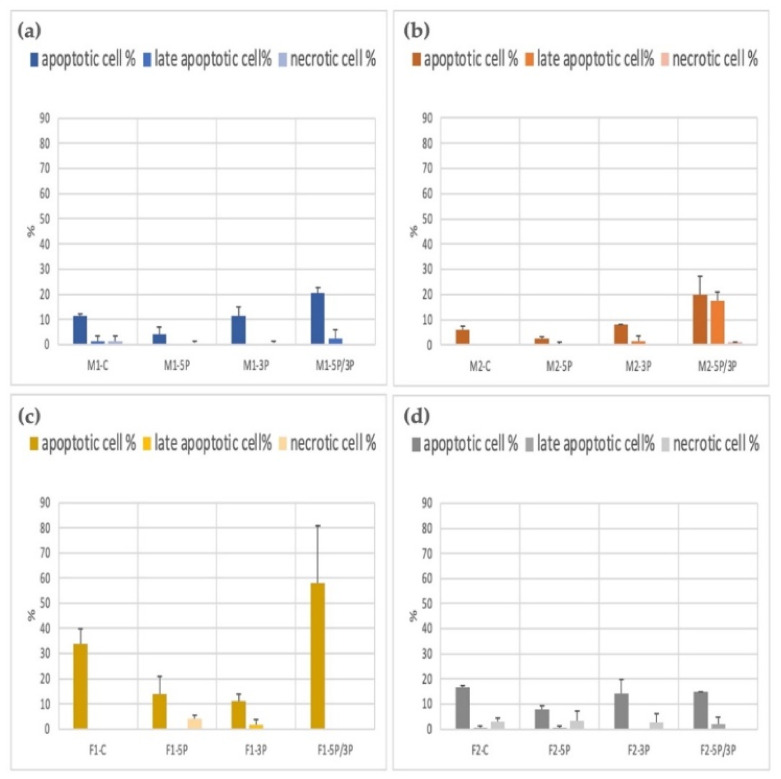
Comparison of the apoptotic, late apoptotic, and necrotic cell percentage in M1 (**a**), M2 (**b**) (ZZ genotype) and F1 (**c**), F2 (**d**) (ZW genotype) PGC lines.: C: non-inhibited, 5P: gga-miR-302b-5P inhibition, 3P: gga-miR-302b-3P inhibition, 5P/3P: gga-miR-302b-5P andgga-miR-302b-3P inhibition.

**Table 1 genes-13-00082-t001:** Primers used at the mRNA analysis.

Gene Symbol	Gene Full Name (*Organism*)	NCBI Number	Primers		Length of the Product (bp)
*GAPDH*	Glyceraldehyde-3-phosphate dehydrogenase (*Gallus gallus*)	NM_204305.1	FW	GACGTGCAGCAGGAACACTA	112
			RV	CTTGGACTTTGCCAGAGAGG	
*POUV*	POU domain class 5 transcription factor 3 (*Pou5f3*) (*Gallus gallus*)	NM_001110178.1	FW	GAGGCAGAGAACACGGACAA	109
			RV	TTCCCTTCACGTTGGTCTCG	
*CVH*	DEAD-box helicase 4 (*DDX4*) (*Gallus gallus*)	NM_204708.1	FW	GAACCTACCATCCACCAGCA	113
			RV	ATGCTACCGAAGTTGCCACA	

**Table 2 genes-13-00082-t002:** Primers used at the miRNA analysis.

Name	Gene	Accession Number	Assay ID	Sequence
miR-92	hsa-miR-92	MI0000719	000430	UAUUGCACUUGUCCCGGCCUG
gga-miR-302b-3P	hsa-miR-302b	MI0000772	000531	UAAGUGCUUCCAUGUUUUAGUAG
gga-miR-302b-5P	gga-miR-302b*	MI0003700	008131_mat	ACUUUAACAUGGAGGUGCUUUCU

**Table 3 genes-13-00082-t003:** Inhibitors used at the miRNA inhibition assays.

Name	Gene	Catalog Number	Assay ID	Type of Inhibitor
anti-miR-302b-3p	hsa-miR-302b-3P	AM17000	AM10081	anti-miRTM-miRNA inhibitor
anti-miR-302b-5p	gga-miR-302b-5P	4464084	MH11349	mirVanaTM miRNA Inhibitor

## Data Availability

The data presented in this study are available on request from the corresponding author.

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
