# Peer review of "Effect of miR-302b MicroRNA Inhibition on Chicken Primordial Germ Cell Proliferation and Apoptosis Rate"

_genes, 2021, doi:10.3390/genes13010082_

Round 1
Reviewer 1 Report
I believe that the paper should be rejected from publication due to plagiarism concerns. This is a section of the paper:
"They represent a subclass of small non-coding RNAs (20-24bp) that fine-tune the regulation of gene expression at the post-transcriptional level, influencing both the stability and translation of mRNAs [14]. RNA polymerase II transcribes the miRNAs as pri-miRNAs that undergo subsequent modifications like 3’ end polyadenylation tailing and 5’ end caping. The primary transcripts are cleaved by the Drosha Ribonuclease III enzyme to produce an approximately 70 nt stem-loop precursor miRNA (pre-miRNA) [15], which is later cleaved by the cytoplasmic Dicer ribonuclease to generate the mature miRNA and antisense miRNA star products (5P and 3P)"
This is a section related to miRNA 302b that I found at www.GeneCards.org :
"microRNAs (miRNAs) are short (20-24 nt) non-coding RNAs that are involved in post-transcriptional regulation of gene expression in multicellular organisms by affecting both the stability and translation of mRNAs. miRNAs are transcribed by RNA polymerase II as part of capped and polyadenylated primary transcripts (pri-miRNAs) that can be either protein-coding or non-coding. The primary transcript is cleaved by the Drosha ribonuclease III enzyme to produce an approximately 70-nt stem-loop precursor miRNA (pre-miRNA), which is further cleaved by the cytoplasmic Dicer ribonuclease to generate the mature miRNA and antisense miRNA star (miRNA*) products."
I am not policing plagiarism and I found that by chance just by fact-checking some of the authors claims regarding miR 302b. I believe this is unacceptable and I cannot not to think that other parts of the paper are reworded from other papers or websites.
Other than this serious issue the following are additional problems that should be addressed:
Introduction: Significant information is lacking regarding known roles of miR 302b.
Methods/Results: The authors list two different types of miR inhibitors that have their own respective negative controls. It is unclear how the experiment is controlled. It seems that 3P and 5P were inhibited using two different assays but then these two are combined during the 3P/5P inhibition. Which negative control is used for the "control" group? For example, if the 3P is used, then the 5P and the 3P/5P are not controlled properly. From the available information it appears to me that this experiment does not cover/control all potential experimental designs. In addition, the fact that the authors detect 50% (!) apoptotic rate for one of their control conditions and still there is a detectable rate of proliferation (and it is not negative since half of the cells are dying(?)) is indicating that there is something not computed correctly in the experiment.
My advice is for the authors to re evaluate this project as a whole.
Author Response
"Please see the attachment."

Reviewer 2 Report
A well written and very interesting manuscript.
Style: "Expression of CVH and cPOUV were measured."
line 201 please this sentence.
Gene symbols: delete c before gene symbol (Table 1, and so on)
Some figures are very small and it is hard to see any details.
Discussion needs some improvements.
It is not clear whether other miRNAs may inhibit PGCs. Please extend what has been reported and discuss previous reports along with your findings.
Line 321: It was published.., please rephrase.
Line 325: .. consequences between cells inhibited ... what do you mean, please clarify.
Line 330: "The double-stranded RNA binding protein may affect the 5P/3P strand selec-330 tion of concurrent expression in cancer cells." do not understand the justification for this.
Line 334 "Other source points" please check wording
Line 334-341: work out the connection to your results. Not clear what you want tell the reader.
Line 350-354: please add references here.
Line 363: reference is missing
Line 365: "The literature has cited ..." please check wording
Line 367: reference is missing
Figure 10: may be placed at the beginning of discussion
Author Response
"Please see the attachment."

Round 2
Reviewer 1 Report
Authors provided some explanation regarding controls for the experiments presented. The experiment as presented is not properly controlled and I highly encourage authors to improve the quality of their experiments by incorporating all controls at the same time. If I was doing this experiment I would include both negative controls separately and combined in addition to "nothing added" control for each of the cultures tested. I am being very lenient in this regard for this manuscript.